# Factors Determining the Involvement in Non-Religious Activities in the Parish: A Cross-Sectional Study of the Catholic Laity

**Krzysztof Jurek \***, **Jadwiga Plewko and Małgorzata Szyszka**

The Institute of Sociological Sciences, The John Paul II Catholic University of Lublin, 20-950 Lublin, Poland; jadwiga.plewko@kul.pl (J.P.); malgorzata.szyszka@kul.pl (M.S.)
**\*** Correspondence: kjurek@interia.eu

**Abstract:** Charitable activities of the Catholic Church in Poland are carried out primarily at two levels: national church organizations, diocesan and religious, and at the level of less formalized parish organizations. The data show a relatively low percentage of people who are strongly involved in parish activities and in non-religious (social, charitable) affairs of the parish community. The first purpose of this paper is to indicate the socio-demographic features that characterize people who are socially engaged. The second aim is to search for model regularities indicating determinants of social activity of parishioners. We conducted the research in parishes of Lublin Archdiocese in 2020. The research sample was 1867 people, of whom 70% were women. The average age of the respondents was 54.31 years. We have selected predictors that characterise the participants of the non-religious activities in the parish. Referring to the theoretical model of social participation and the concept of social capital, we have indicated the factors that shape the pro-social attitudes of parish members.

**Keywords:** community engagement; parish; social participation; determinants of community engagement

## 1. Introduction

The Polish society is still characterized by one of the highest rates of religiosity in Europe. As of 2021, 91.9% of Poles declare themselves as Catholic, and 32.5 million believers belong to the Catholic Church in Poland (Przeciszewski 2021). The Catholic Church, as an institution, has 15 metropolises, which include 45 dioceses, and is divided into 10,382 diocesan and religious parishes (Kasper 2021). The mission of the universal Church includes worship and works of evangelization, as well as the performance of works of mercy. Caritas, as Pope Benedict XVI emphasizes, belongs to the nature of the Church; it is an inalienable expression of her essence (Benedykt XVI 2006). The Catholic Church in Poland is also very committed to charitable functions and is the largest organization, after the state, that provides assistance to the needy. This is confirmed by data from the Central Statistical Office, which show that socio-religious entities more often than other third sector organisations declare activities in the field of social and humanitarian assistance. (Kamiński 2019). Charitable activities of the Catholic Church in Poland are carried out primarily at two levels: national church organizations, diocesan and religious, and at the level of less formalized parish organizations (Sadłoń 2021). The Church's charitable activities, not only in the colloquial sense but also in official documents, are often called charitable, social, or often non-religious activities.

A significant part of aid activities is carried out locally, in parishes and in relation to the needs of the local community (Biela 2021). This is expressed both in the statutory activities of parish associations and organisations and in various forms of grass-roots self-help initiatives, which are a manifestation of the interpersonal solidarity of Catholics. The most important tasks that a parish may carry out as an ecclesiastical unit, which at the same time

has legal personality under the state law, are educational activities, charitable activities, protection and support of the family, protection of cultural goods and national heritage, cultural development, health care and development of physical culture (Steczkowski 2009). All dimensions of parish activity require the cooperation of the parish priest, other clergies, and lay Catholics, who have their legal status in the Church and specific tasks to fulfil. This creates the possibility for the active participation of the faithful in the life of the parish, and at the same time opens the way for Catholics to take responsibility for their communities of faith in the spirit of the ongoing process of the search by lay Catholics for their own subjectivity in the Polish Church. The involvement of the faithful in organizational activities within the parish is an important condition for the realization of these tasks. The effects may be both a strengthening of their faith and religiousness, as well as concrete assistance provided to those in need.

The Catholic Church in Poland includes numerous organizations of the laity that operate in parishes. According to the research conducted by the Institute for Catholic Church Statistics (ISKK), there are currently 65,500 more or less formalized parish organizations (Przeciszewski 2021). According to a survey of parish organizations regularly conducted by ISKK, almost 52% of all such communities are involved in charitable activities. The peculiarity of Polish Catholicism is the strong identity, bonds and activity of people who engage in small prayer, and formation and charity communities (Petrowa-Wasilewicz 2021). Parishes are one of the important centres of community life in the local environment. Members of parish organizations, to varying degrees (depending on the type of organization, the quality of formation and the degree of personal involvement), internalize and disseminate values such as solidarity, the common good, trust, participation, cooperation, subjectivity and openness. The conviction of the importance of these values belongs to the civic attitude and determines the appearance of civic behaviour (Wnuk-Lipiński 2005). The parish, as the primary form of the Church's presence in society, shapes a certain type of social personality of its members (Świątkiewicz 2010). The issue of the significance of lay activity within the parish seems to be an important counterbalance to the less favorable phenomenon of the relatively small number of the faithful involved in various forms of church, including parish, activities. Against the background of the very large number of Catholics belonging to the Church, the percentage of those actively involved in activities on behalf of church communities ranks low. Currently, less than every 10th Catholic is seriously involved in the organizational life of the Church. The low percentage of active persons reflects the generally low level of social activity among Poles, although indicators of involvement are slowly rising. This phenomenon is clarified by Dixon, who states that the level of parishioners' involvement decreases when there are more Catholics in the parish. This can be explained by the fact that people may show less willingness to take an active role in parish life if they think that there is a great deal of other people around who are able to take the initiative for them (B. Dixon 2010). According to national surveys conducted every two years since 1998 by the Centre for Public Opinion Research (CBOS), social work in civic organizations was initially declared by 23% of respondents, while currently (data from 2020) it is declared by 43% of surveyed Poles (CBOS 2020). In the case of the Catholic Church faithful, about 8% (2018), which is about 2.5 million people are involved. According to data from February 2020, Poles are slightly less likely than two years earlier to declare that they devote their free time to activities in organizations, religious movements, church, and parish communities; there has been a decline of 1.4 points. When analysing the data on the relatively small number of parishioners directly and actively involved in work in parish communities, it should be remembered that until 1989, Catholic parishes in Poland were limited in conducting non-cult activities, and parishioners did not have much experience in organizing social activities within their parishes. It takes some time to become active, to break the passive attitude and to develop new forms (Mariański 2020). Parishes fulfil primarily sacral (religious) functions, but they are also a place for building social ties, creating cultural and educational initiatives, and even sometimes economic ones. Small religious communities have non-religious functions,

because the parish is a religious and social institution and, in part, activities in these two dimensions overlap. Sociologists point to social and humanizing functions; social problems that interest everyone, or almost everyone, and must be solved together. Of the more than 65,000 parish organizations surveyed by the Institute for Catholic Church Statistics, the largest number are directed toward children and youth, followed by the elderly, the poor, the disabled, the unemployed, and victims of violence. Sociological literature, ascribing to the parish the value of multifunctionality, lists the following non-religious functions: care and protection, cultural and educational, administrative and economic, recreational, tourist and pilgrimage, and advisory and intervention (Świątkiewicz 2010).

The purpose of this paper is primarily to identify forms of parishioner involvement in non-religious activities in the parish. The article is an attempt to explain the socio-demographic characteristics of parishioners, which determine the assessment of their influence on non-religious (social, charitable) activity in parishes. A literature review on this subject shows a lack of research on the non-religious activities of parishioners. Therefore, undertaking such an analysis is of significant exploratory importance. The second aim of the study of social involvement of people functioning within one of the smallest types of local communities, the parish, is to search for model regularities indicating the determinants of social activity of parishioners. Non-profit activity, oriented towards the realization of interests and values of various social groups on different levels: social, cultural, economic, civic, political, is identified with the process of social participation (Jabłoński and Szymczak 2021). Therefore, we refer here to the basis of the theoretical model of social participation, applying its assumptions to determine the factors of shaping pro-social attitudes of members of the local community. According to the concept of social participation seen as a person's experience, the process of social involvement has four dimensions: 1. subjective justification of participation (indicators include: conscious decision of a given person, goal orientation, autonomy and voluntariness of involvement, definition of one's role and tasks, sense of responsibility); 2. axiological content (indicators: other person as a value, moral, social, civic, aesthetic, self-realization, religious values); 3. sense of agency (indicators: sense of influence on reality and effectiveness in action, sense of influence on social issues); 4. experience of participation (indicators: experiencing the value of the community, recognizing oneself as a subject in the community, orientation towards common good, experiencing bidirectionality and reciprocity, i.e., contribution of the person and receiving from the community). Referring to these assumptions and the results of own research, we propose a scheme of determinants for social involvement of parishioners (Szymczak 2013).

Social commitment, which entails networks of connections between individuals and soft skills, aiming to fulfil the common needs or matters, is one of the fundamental elements of social capital. Social capital is defined in various ways in the sociological literature. Hereby, various aspects of this phenomenon are emphasised here and are to be found in the three most classical approaches to this idea. Pierre Bourdieu perceives social capital as own resource of each individual rather than a community (Bourdieu 1983) but related to a network of interpersonal relations. According to Robert Putnam, social capital is an attribute of a community, especially in relation to the social benefits of its existence. Another author, James Coleman, indicates a group (social structure) as the creator and main beneficiary of social capital (Coleman 1988). In this context, it is worth paying attention to the interest of some researchers in religious structures as a place for creating and developing social capital, supporting the activity and social commitment of the believers, maintaining norms, and building relationships. This is especially emphasized by Putnam's research conducted in the United States, which shows the positive relationship between social capital and religiosity. Faith-based communities constitute an important reservoir of social capital there (Putnam 2000) and are the most frequently indicated places for creating voluntary attitudes and activities (Becker and Dhingra 2001). According to Putnam, religious groups support social activity and are an incubator of social skills, norms, or social commitment (Putnam 1994). Numerous social studies conducted in various countries

have mainly focused on the approach to religion (faith) as a source of social capital of religious communities (Candland 2000; R. E. Dixon 2010; Oreshina et al. 2015; Williams and de Mola 2007; Wuthnow 2002). The research presented by us shifts the emphasis of interest to important, but other than religiousness and less frequently analysed component of parish life, i.e., its non-religious dimension of activity, its extensive functions. Their accomplishment would not be possible without the existence in the parish environment of such features as bonds of trust, loyalty, solidarity, cooperation for the common good, i.e., classic manifestations of social capital: real and potential resources embedded in the network of relations by an individual or group, made available through them and derived from them (Nahapiet and Ghosal 2000). Parishes provide a stable and safe environment for the development of various types of social commitment (Cassel 1999). The issues of social and non-religious activities of parishioners as a research area is therefore an extension of the current perspective of researching the social capital of religious structures.

It should be emphasized that the empirical research presented in this paper is not representative. Therefore, the analyses and conclusions should be treated as a pilot study. Nevertheless, it is interesting and worth presenting as a contribution to further research.

## 2. Materials and Methods

### 2.1. Research Design

The parishes of the Archdiocese of Lublin were included in the study. The bishopric of Lublin was created in 1805 on the basis of a bull issued by Pope Pius VII. The diocese of Lublin covered an area of 23,250 km$^2$ and had 330 thousand people. It was divided into 17 deaneries and 214 parishes. Major changes in the diocese of Lublin began to take place during the Second Republic of Poland. The outdated parish network was supplemented by creating about 100 new parishes, the pastoral ministry was reorganized and streamlined, and a number of church institutions, such as charitable institutions and schools, were established. Not without significance is also the fact that in 1918 the Catholic University of Lublin was founded, with the Bishop of Lublin as its the Great Chancellor. In 1992, the administrative structures of the Church in Poland were reorganized. By virtue of the bull "Totus Tuus Poloniae Populus", Lublin diocese was raised to the rank of archdiocese and a metropolis was established in Lublin. Currently the Archdiocese of Lublin covers an area of 9108 km$^2$, which is inhabited by approximately 1.75 thousand people, of which over a million are Catholics. There are 271 parishes in the archdiocese. There are about 900 diocesan priests who serve in various pastoral capacities in the diocese and beyond its borders, either working in teaching and research, working in diocesan institutions, or who are retired. In addition, the Archdiocesan community includes nearly 200 priests, over 500 nuns, and nearly 50 alumni preparing for priestly ordination. Of the more than one million Catholics—residents of the Archdiocese of Lublin— 38% attend church and 20% receive Holy Communion (which is 53% of those attending Mass on the day of the count). Of those attending Sunday Eucharist, 40% are men and 60% are women. On the other hand, among the communicantes, 34% are men and 66% are women (Data of the Archdiocese of Lublin 2018). These data reflect nationwide rates of dominicantes and communicantes. In 2019, there were 36.9% and 16.9%, respectively (Annuarium Statisticum Ecclesiae in Polonia 2020). In this aspect, the Archdiocese of Lublin can be treated as a "picture" of statistical average.

When considering the parishes of the Archdiocese of Lublin as a subpopulation of the parish population in Poland, it is necessary to point out the characteristic features resulting from the geographical location and economic and cultural conditions of the Lubelskie voivodeship (although the area of the Archdiocese is larger than the province). In general, Lubelskie is an agricultural region with a relatively high unemployment rate (8.2%, Poland—6.2%), an average gross salary lower than the national one (4300 PLN, Poland—5100 PLN) and a much higher percentage of poor households than the national average (income poverty in 2018 affected 26% of households in Lubelskie voivodeship, average for Poland—13%). The percentage of people satisfied with their living conditions

in Lubelskie is one of the lowest in the country (in 2018 it was 59%, the average for Poland—63%), thus Lubelskie is one of the regions with the lowest level of satisfaction with life (76%, average for Poland—83%). At the same time, it is the region with the highest percentage of believers and deep believers (91%, Poland—81%). It ranks 9th among Polish provinces in the number of marriages, 8th in the number of marriages dissolved by divorce, and 11th in the number of separations pronounced (Demographic Yearbook 2020).

### 2.2. Participants and Procedure

The analyses included in this article are part of the study conducted by the Institute of Sociological Sciences in the XXX on the perception of parishes as a source of social support. The research was conducted in January and February 2020 in the parishes of the Lublin Archdiocese. The survey covered the entire area of the Archdiocese. Twenty questionnaires were distributed to parishioners in each parish, and the criteria for selecting respondents included the following categories of people or groups: families with children, single parents, elderly people (over 60), single people, dependent people (including people with disabilities). The data collection technique was a survey questionnaire distributed for self-completion by respondents. The method of distribution of the questionnaires involved the delivery of questionnaires by internal mail to each parish, which were then distributed by the parish priest(s) to parishioners according to the criteria adopted for the selection of respondents. After completing the questionnaires, parishioners returned them in sealed envelopes to the parish office, which were then sent back to the XXX. In the end, 150 parishes returned the questionnaires (55.3% of the parishes that returned the questionnaires). A total of 1931 questionnaires were received (36% return rate) and 1867 completed questionnaires qualified for analysis. The survey return rate is similar to surveys conducted in other archdioceses (e.g., the Archdiocese of Łódź) (Kaźmierska 2018).

It should be clearly emphasized that, due to the sample selection and the method of collecting the data, the research sample is not representative and does not correspond to the distribution of socio-demographic characteristics in the society. Similar problems were faced by American researchers, among others the team of D. C. Leege (Leege and Trozzolo 1985), and Centre for Applied Research in the Apostolate (CARA) (Zech et al. 2017). Therefore, we stress caution in generalizing from our findings.

The procedure was approved by the Research Ethics Committee at the Institute of Sociological Sciences of the John Paul II Catholic University of Lublin (protocol code 11/DKE/NS/2021).

### 2.3. Statistical Methods

The analysis began by characterising the study group. The percentage and frequency of each category was indicated. Descriptive statistics (means, standard deviations) were calculated for the quantitate variables. The Chi-square test of independence was used to compare forms of engagement between men and women. CATREG was used to analyse the determinants of parishioners' social involvement, while normal regression analysis was used to examine models with only interval variables; categorical regression makes it possible to analyse models with nominal, ordinal and interval variables. In CATREG, nominal variables are quantified by assigning numerical values to the categories. After rescaling, nominal and ordinal variables are treated as interval in the procedure to find an optimal linear regression equation.

## 3. Results

The research sample was 1867 persons—parishioners of the Archdiocese of Lublin, of whom 70% are women. The average age of the respondents was 54.31 years. Nearly 40% of the parishioners surveyed have a college degree, more than one-third have a high school education and 58% of the respondents are married. Of the 80% who have children, more than one-third have two children, one in five have three children, while 14% are raising one child. The vast majority of respondents rate their financial situation well. In terms

of Mass attendance, two-thirds of respondents report regular weekly attendance. 38% of parishioners belong to religious communities, associations, or ministries, and 19% perform some function in them (Table 1).

**Table 1.** Characteristics of the study group.

| Characteristics | Categories | Parameter | |
| --- | --- | --- | --- |
| | | N/M | %/SD |
| Gender | Female | 1305 | 69.9 |
| | Male | 538 | 28.8 |
| | NA | 24 | 1.3 |
| Age | | 54.31 | 16.49 |
| Children | Yes | 1478 | 79.2 |
| | No | 389 | 22.8 |
| Marital status | Married | 1075 | 57.6 |
| | Single | 769 | 41.2 |
| | NA | 23 | 1.2 |
| Number of children | 1 | 265 | 14.2 |
| | 2 | 660 | 35.4 |
| | 3 | 360 | 19.3 |
| | 4 and more | 168 | 9.0 |
| | NA | 25 | 1.3 |
| | No children | 389 | 20.8 |
| Education | Elementary | 143 | 7.7 |
| | Vocational | 307 | 16.4 |
| | Secondary | 672 | 36.0 |
| | Higher | 723 | 38.7 |
| | NA | 22 | 1.2 |
| Attendance at Mass | Several times a week | 485 | 26.0 |
| | Once a week | 1155 | 61.9 |
| | 1–2 times per months | 111 | 5.9 |
| | Several times a year | 76 | 4.1 |
| | Not at all | 12 | 0.6 |
| | NA | 28 | 1.5 |
| Material situation | Very good | 158 | 8.5 |
| | Rather good | 712 | 38.1 |
| | Satisfacory | 885 | 47.4 |
| | Rather bad | 87 | 4.7 |
| | Very bad | 6 | .3 |
| | NA | 19 | 1.0 |
| Membership in religious communities, ministries or organizations in the parish | Yes | 712 | 38.1 |
| | No | 1125 | 60.3 |
| | NA | 30 | 1.6 |
| Functions held in parish organizations | Yes | 352 | 18.9 |
| | No | 1354 | 72.5 |
| | NA | 161 | 8.6 |

N—frequency; %—percentage; M—mean; SD—standard deviation.

### 3.1. Non-Religious Involvement of Parishioners

There were 1082 in the women's group who were involved in at least one form of activity. In the male group, there were 461 such individuals. The most common forms of involvement in non-religious life of the parish include work around the church, such as cleaning and decorating. A significant percentage of respondents were involved in collections for the needs of the church, collecting food or clothing for the needy, and helping the elderly, lonely and disabled. There were no differences between men and women in

these activities. Other forms of involvement appeared less frequently. Differences between men and women were noted for activities such as building and maintenance work and fundraising for parishioners ($p < 0.05$) (Table 2).

**Table 2.** The most popular forms of parishioner engagement.

| Forms | Female [N = 1082] | | Male [N = 461] | | Statistics |
|---|---|---|---|---|---|
| | N | % | N | % | $\chi^2$ |
| Work around the church (cleaning, decorating) | 815 | 75.3 | 328 | 71.1 | 0.886 |
| Collecting money for the church | 449 | 41.5 | 211 | 45.8 | 2.992 |
| Construction, maintenance | 55 | 5.1 | 117 | 25.4 | 134.430 *** |
| Collection of money for the needs of parishioners | 203 | 18.8 | 115 | 24.9 | 7.983 ** |
| Collection of food and/or clothing for the needy | 311 | 28.7 | 125 | 27.1 | 0.136 |
| Counselling for those in need | 61 | 5.6 | 24 | 5.2 | 0.019 |
| Organization of recreation for children and youth | 45 | 4.2 | 25 | 5.4 | 1.069 |
| Help the elderly, lonely, disabled | 209 | 19.3 | 99 | 21.5 | 1.127 |
| Editing the parish newspaper | 27 | 2.5 | 13 | 2.8 | 0.062 |
| Organizing cultural and sporting events | 75 | 6.9 | 40 | 8.7 | 1.404 |
| Organizing activities for children and youth | 50 | 4.6 | 31 | 6.7 | 2.731 |
| Organizing activities for the elderly | 22 | 2.0 | 12 | 2.6 | 0.315 |

$\chi^2$—chi–squere test; ** $p < 0.01$; *** $p < 0.001$.

*3.2. Socio-Demographic Characteristics of Parishioners with a Sense of Influence on the Non-Religious Life of the Parish*

The CATREG analysis identified those socio-demographic variables that are associated with the belief of influence or lack of influence on participation in the organization of the parish's non-religious activities (F = 16.251; $p < 0.001$). Significant predictors were found to be gender ($p = 0.016$), marital status ($p = 0.005$), education ($p < 0.001$), assessment of financial situation ($p = 0.002$), attendance at Mass ($p = 0.005$), holding a function ($p = 0.001$), tie with parish ($p < 0.001$), age ($p < 0.001$), number of children ($p = 0.001$), interest in the nonreligious life of the parish ($p = 0.001$), use of parish-organized instrumental help ($p = 0.009$), and assessment of material situation ($p < 0.001$). The model explains 25% of the variation in the dependent variable. Table 3 shows the quantification values obtained by applying the alternating least squares (ALS) algorithm. Impact coefficients were calculated to determine the relationship of the predictor categories to the dependent variable. The highest sense of influence was reported for parishioners expressing a high interest in parish affairs, holding office with parish organizations, and receiving instrumental assistance organized at the parish.

**Table 3.** Results of qualitative regression analysis (CATREG).

| Variables | Categories of Variables | Quantification | Beta | Impact Coefficients |
|---|---|---|---|---|
| Gender | Female | −0.586 | 0.066 | −0.039 |
| | Male | 1.706 | | 0.113 |
| Marital status | Married | 0.676 | 0.094 | 0.064 |
| | Single | −1.479 | | −0.139 |
| Education | Elementary/vocational | −1.256 | 0.107 | −0.134 |
| | Secondary | −0.448 | | −0.048 |
| | Higher | 1.192 | | 0.128 |
| Material situation | Good | 0.881 | 0.066 | 0.058 |
| | Avarage | −1.045 | | −0.069 |
| | Bad | 1.531 | | 0.101 |
| Attendance at Mass | Once a week | −0.858 | 0.049 | −0.042 |
| | Several times a week | 0.471 | | 0.023 |
| | Less often | −0.048 | | −0.002 |
| | Not at all | −11.143 | | −0.546 |
| Functions performed in parish organizations | Yes | 1.924 | 0.110 | 0.212 |
| | No | −0.520 | | −0.057 |
| Tie with parish | Yes | 0.242 | 0.095 | 0.023 |
| | No | −5.057 | | −0.480 |
| | Hard to say | −3.406 | | −0.324 |
| Age | =<40 | 1.295 | 0.076 | 0.098 |
| | 41–50 | −1.446 | | −0.110 |
| | 51–60 | 0.500 | | 0.038 |
| | 61–70 | −0.638 | | −0.048 |
| | >70 | 0.924 | | 0.070 |
| Number of children | 1 | −1.894 | 0.055 | −0.104 |
| | 2 | 0.525 | | 0.029 |
| | 3 | 0.788 | | 0.043 |
| | 4 and more | −0.713 | | −0.039 |
| Interest in non-religious life of the parish | Low | −1.420 | 0.395 | −0.561 |
| | Average | −0.113 | | −0.045 |
| | High | 1.129 | | 0.446 |
| Housing situation | I live alone | 2.509 | 0.059 | 0.148 |
| | I live with my family | −0.399 | | −0.024 |
| Use of instrumental help organized in the parish | Yes | 2.140 | 0.078 | 0.167 |
| | No | −0.467 | | −0.036 |
| $R^2 = 0.268$; Adjusted $R^2 = 0.251$ | | | | |

## 4. Discussion

The purpose of this study was an attempt to identify social and demographic variables that are related to parishioners' evaluations in order to assess their impact on the non-religious life of the parish. The results of the study should be of particular interest to the church authorities/leaders.

They can better recognise and identify the categories of parishioners who have a sense of influence on the functioning of the parish community. This knowledge can therefore be useful in taking action to engage and promote parishioner involvement. It seems that, in the context of secularisation processes, sociological knowledge about the socio-demographic profile of parishioners with a sense of influence on parish activities (parishioners involved in non-religious life) is very interesting. Participation in the decision-making process increases parishioner morale, satisfaction, and involvement in parish life (Conrad 1988). Pargament (2002) noted that not everyone experiences the same benefits from religion.

Religiosity is more helpful for more socially marginalised groups (e.g., the elderly, the poor) and for those who are more committed.

Our research has shown that parishioners who express a strong interest in parish events and hold positions in parish organisations have a stronger sense of influence over the non-religious life of the parish. In line with the idea of social participation, such individuals have a greater sense of agency. Participation in official structures enables them to take part in decision-making. In this sense, involvement in parish activities values the individual. A study conducted in the USA in 2002 found that both men and women feel more valued for their participation in church if they are currently involved in various parish activities and if they are elected to the church board. In our study, we found that men and those aged under 40 had a greater sense of influence over non-religious activities in the parish. Lummis (2001) argues, in turn, that younger men feel more appreciated than older men, even if they do relatively little in church, while women's age is unrelated to their sense of being valued for their participation (2004). Those who received instrumental assistance organized in the parish had a greater sense of influence on non-religious activities. Perceived spiritual and social benefits have a positive and significant relationship with church participation among regular and irregular church goers (Casidy and Tsarenko 2014). According to the rational choice theory of religion, participation in religious/church activities is determined by a rational assessment of the potential costs connected with that participation (e.g., time, effort) and the potential benefits (such as spiritual, social, instrumental intellectual, entertainment support) (Christiano et al. 2008).

The parish thrives on the commitment of the faithful, on the one hand by building up a community of faith (strictly religious activities), and on the other by non-religious activities that serve the local community, such as social and psychological support. It seems that those who attend Mass regularly and frequently feel more influenced by the non-religious activities of the parish. The literature shows that believers who are more religiously committed perceive more benefits from participating in church activities, both religious and non-religious (Iannacone 1992). Cieslak (1984) indicates that overall parishioner involvement is related to financial donations to the parish, weekend Mass attendance, and involvement in non-liturgical parish activities. Small and medium-sized parishes that exhibit "responsiveness" are shown to have higher parishioner involvement. This relationship is much weaker for large parishes. Perhaps small and medium-sized parishes provide opportunities for clergy to build deeper relationships with parishioners.

In our study, we have shown that parishioners who feel a connection to the parish are characterised by greater involvement in non-religious activities. Ploch and Hastings demonstrate that parishioners' ties/close relationships (friendship network) are a stronger predictor of church attendance than parental attendance (socialization) or family status. Strong bonds between believers foster community building, identify, and create various forms of mutual support (Ploch and Hastings 1998). American research has shown changes in the social commitment of parishioners. In order to adopt common interests, they are seeking to participate in common meetings (e.g., retreats) to provide services to those in need (Sweetser and Forster 2011). Analyses thus indicate the increasing role of the laity in parish decision-making (Zech et al. 2017).

The second aim of the analysis was to inscribe the results of the research in the concept of social participation and an attempt to indicate which factors have a key influence on the activity of community members and the building of social bonds, and in a broader perspective on the formation of pro-social attitudes of parish members. The starting point is the model of social participation (Szymczak 2013) as an interesting source of inspiration for the author's concept of factors determining the social involvement of the faithful for their own parishes, perceived as local communities. The parish is thus treated as a local community. Adopting the interaction-processual perspective (Kaufman 1959; Turowski 1999; Pietraszko-Furmanek 2021), a parish is a community functioning in a separate space, whose members are linked by a sense of identity, solidarity, and shared values, who communicate with each other and undertake grassroots initiatives (social actions) pursuing

a common interest, thus building a social bond. Social participation is understood as a person's involvement in activities that provide interaction with others in society or the community (Levasseur et al. 2010). Therefore, in conceptualising the determinants of parishioners' social engagement, it is important to consider both the wider environmental, social, and cultural context and the personal characteristics of those in social roles and their preferences. (Piškur et al. 2014).

Considering the above-mentioned findings, a scheme for social participation in the parish, commonly understood as a local community, was developed, with particular reference to the interdependence of factors determining the social involvement of its members and actors. The analyses in the first part of this article identified the determinants of parishioners' social engagement as community members, i.e., their individual metric characteristics. Twelve significant predictors defining a socially engaged parishioner were identified. The conformity of the results of our own research with the model of social participation is evidenced by, among others, the occurrence of a sense of agency and influence on the social, non-religious life of the parish (the second dimension of the model) among parishioners declaring a high interest in parish affairs, as well as performing functions in parish organisations. The study of parishioners revealed the dependence of their sense of influence on social life on certain socio-demographic variables. The common elements of the model of social participation and our own research also relate to the role of an axiological content (especially religious, moral, social values and an attitude to help other people) as the basis for the non-religious activity of the people surveyed (the third dimension of the model). Some other converging single indicators can also be mentioned, such as voluntariness of involvement (one of the indicators of the first dimension), orientation to the common good of the community, and the experience of community values (as indicators of the fourth dimension of the model) (Kasper 1986).

The obtained results respond to a problem of building social capital based on the parish and its resources. Non-religious involvement is strongly related to the social capital built around the parish. As Dixon points out, social capital that grows within the parish structure has certain attributes (trust, commitment of parishioners, common goal, etc.), is generated and coordinated by leaders, has both internal (bonding) as well as external (bridging) features. The bonding social capital can be thought of as the networks, trust, beliefs, and behaviours which create and build up the parish community. The bridging social capital is created when the parish lives out its mission through outreach to the wider community (B. Dixon 2010). The parish affects the external environment, which includes its institutional links to the diocese and the universal Church, as well as to the national and Catholic culture in general. Dixon emphasises that another important aspect of building social capital is the demography of the local Catholic population (e.g., gender, age, marital status) and social factors (e.g., functions in the parish, leadership) (2010). For example, consolidating the position of a leader or fulfilling functions in parish organizations constitutes a very strong predictor of commitment to the life of the parish. If leaders encourage parishioners to use their talents and skills, it is highly likely that they would be more involved in parish issues. Leege's research shows that parish leaders are more oriented to the performance of their parish as a religious and social community. For them, community is a "property of the particular,", i.e., it is developed through the actions of the local church (Leege 1987). Participation in parish life for many people makes an important form of building social capital. The local parish is one of the most important places where citizens often meet with each other, build interpersonal relationships, a network of formal and informal contacts, spend their free time, look for financial, instrumental, or informational support. In this sense, the parish can be understood as a local community. Parish as community is not a product solely of friendliness but of the feeling of rootedness. Thus, the community is proclaimed by the way it is lived (Leege 1987). In our research, we have shown that people who feel a strong bond with the parish are more involved in non-religious activities for its benefit. Nevertheless, Dixon suggests that an important role in building the social capital of the parish is also played by people who are not involved in the life of the parish, or

their commitment is low. A parish that relies solely on people who have a strong sense of connection with it may endanger the risk of being isolated from the rest of society (B. Dixon 2010). In this context, there is an important role for both priests and lay leaders in undertaking not only religious, but also non-religious activities. This has to inspire parishioners to joint ventures and build a civil society (Wuthnow 1991). Moreover, our research shows that people living alone are more involved in non-religious activities for their parish. Perhaps it can be justified by the fact that parish provides them with a sense of belonging, and at the same time these people, thanks to their commitment, feel more appreciated and valued.

Many researchers prove that active members of the church community (parish) are more motivated, for example, to participate in aid measures (Becker and Dhingra 2001). It is indicated in the literature that the influence of religious involvement on social participation/activity manifests through the social networks of community members (ties, norms, values) and not through the content or strength of religious beliefs (Birdwell and Littler 2012; De Hart and Dekker 2005). Putnam believes that religious people are more likely to volunteer or donate to charity only because of social networks (1994). Research, e.g., in Great Britain, shows that religious people are also more eager to show social commitment, such as volunteering and charity work, than people who declare themselves as non-believers. Another issue is that the strength of parishes (churches) is also to be found in their connections with local communities and the ability to keep them motivated and use local resources, including volunteers (Gallet 2016).

The research presented here focuses mainly on the specific factors that determine (and thus can both stimulate and inhibit) the involvement of the faithful in social activities for the parish. However, social participation is conditioned not only by the personal characteristics of parishioners (individuals). The factors arising from the essence of the parish as an institution are also important. Therefore, in the diagram (Figure 1) we indicate the importance of such elements as: communication between priests and parishioners, effectiveness of information flow, network of connections, level of partnership, awareness of common goal, material resources and human resources (both parish and parishioners), as well as priests' attitude and their openness to the actions of parishioners involved. The aforementioned factors can also play both a role in stimulating the social activity of parish members and, under certain circumstances, not fostering their activation. This opens up the possibility of learning about and shaping those characteristics of the community that may be considered important, valuable and desirable for its development. This is exemplified by the way in which communication takes place between the clergy and the faithful, between the pastor and the leaders of the various communities and associations that function within the parish structure, and between parish members who are not affiliated with small communities.

In the literature and practice of church life, an optimal model of parish management has been developed for some time. One of the basic postulates appearing in the sociological reflection on parishes points to the necessity of rejecting the model of the parish as an institution, an office, or an administrative centre, in favour of becoming communities in which lay Catholics would feel themselves real subjects of the life of the Church, responsible for their parish communities (Mariański 2020). Verification of the new model of the parish, both in the religious dimension, the parish as a community of communities (Chmielewski 1999) and in the process of social interaction of the parish with the local environment, requires careful discernment and highlights the need for the implementation of empirical and comparative research on a wider scale. This postulate should also be applied to other characteristics of the parish such as the local community, in order to have the widest possible knowledge of the potential of parish-type local communities that can be activated in the process of the social participation of its members.

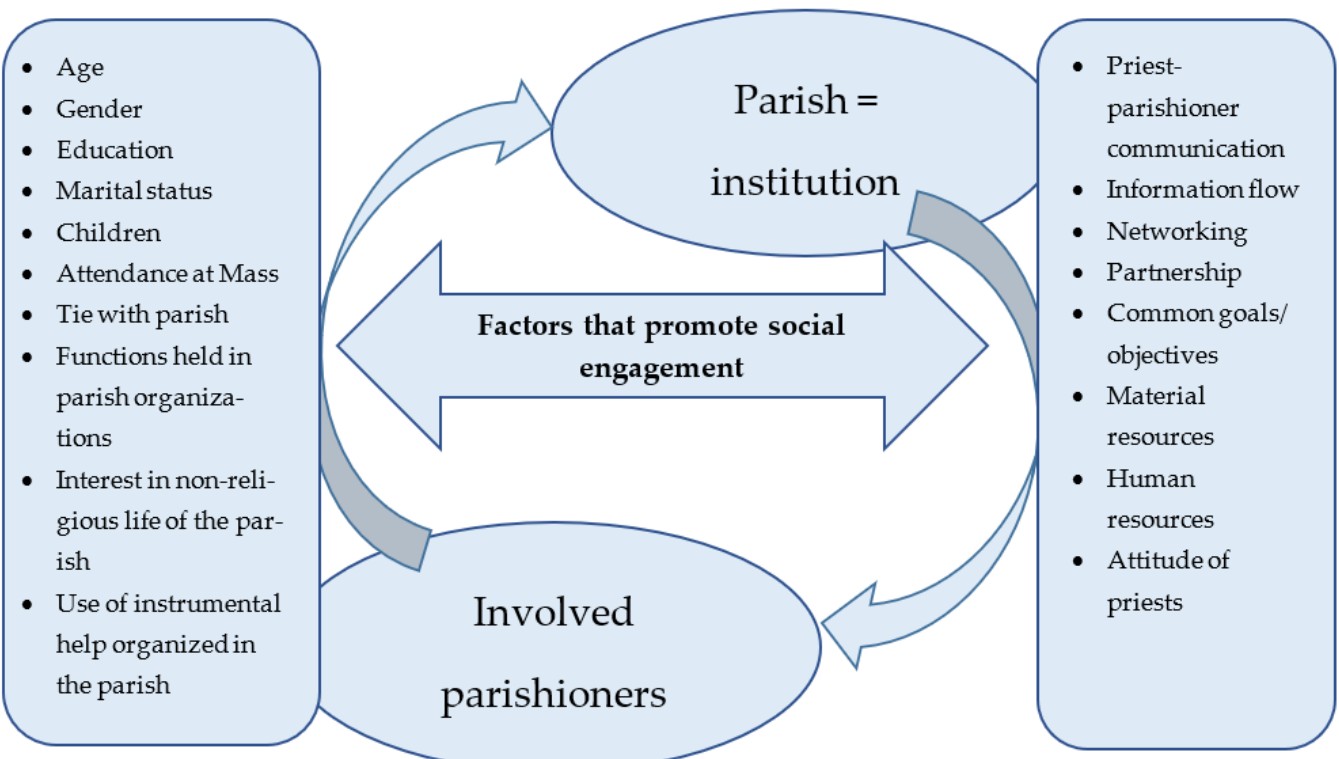

**Figure 1.** Determinants of parishioners' social involvement (perspective of the parish as a local community).

In sociology, since the times of August Comte, the phenomena of social engagement and activities of individuals leading to creating a common community are both conveyed in a following question: What connects, unites, and makes people and social communities stick together? In spite of the fact that the responses of sociologists to the foregoing question differ from each other, they converge on the category of a social bond. According to Robert Nisbet, the aforementioned concept of a social bond appears to be a power, which enables people to stick together in, so called, "social molecules" (Nisbet 1970). Apart from the fundamental religious bond, moral and behavioural ones also play an important role in parish communities; they are based on common activities. Admittedly, not all members of a parish, which is both an institution as well as a community, manifest their bond spirit in the same way. It can be elaborated, as Peter Worsley emphasizes, on existence of two types of bonds in social communities: objective or subjective. The objective ones result from mutual relationships, similarities in lifestyles (e.g., place of residence) and official affiliation to specific groups. In turn, the subjective bonds are based on a sense of belonging shared with other group members and their mutual solidarity (Worsley 1984).

Therefore, the parish is a place of forming bonds and building a community in an objective and subjective sense. Objective sense means that parishes operate in a specific area. Everyone within it has right and opportunity to participate in parish life; naturally, only if one shows such a willingness. Then, a subjective bond refers to a community of values, norms, symbols, rituals, and invariable ceremonies. In this sense, the parish is a place, in which people build their own personal and social identity. What is more, it is also an area where a moral bond based on trust and solidarity is formed. This is expressed within participation in social activities for the parish. In other words, undertaking common religious and non-religious activities is a constitutive element of a parish, which is perceived as a local community (social group). The awareness of one's impact on the broadly understood parish life gives a sense of being an actor (subject), i.e., sense of agency. Parish participation indicates a departure from (extremely institutional) views of the Church, in which only priests are active subjects of pastoral ministry, and the laity representatives

constitute a passive and peripheral "mass" for the parish as a community (Dantis 2016; Mariański 2014).

The research presented in this article did not address the determinants of community engagement that arise from the idea of the parish as a local community. The proposed list is the result of my own reflections based on literature and research and certainly does not include a complete set of possible determinants. However, this points to the need for a dual perspective on the concept of community participation. This approach would reveal the interdependence of community engagement factors. The more factors activating parishioners on the part of the parish, the more active and creative people among parishioners. Conversely, the greater the interest of community members in its non-religious affairs, the greater the awareness of common purpose and the higher the level of partnership.

Thus, the suggested scheme opens a wide spectrum of future directions for social research on the concept of social participation in the local community, of which the parish is a good example. We believe that the issue of benefits derived from participation in various non-religious practices of the parish community should become an area of further scientific research. Apart from the widely understood aid activities, other needs may also occur. For example, Neal Krause and Elena Bastida researched whether church support is related to health (Krause and Bastida 2011). Thus, it is worth extending the investigated problems in order to include the widely understood, extensive functions of parishes. In addition, the direction of further research should concern engagement in non-religious activities, and the creation of communities not only based on primary bonds (primary groups). Other areas of future research may include the following problems: Does parish affiliation constitute a factor in engaging in community activities, or, rather, whether an individuals' personal traits cause such a social engagement? Does religious engagement enhance parishioners' social commitment? In our opinion, providing an answer to the foregoing research questions requires interdisciplinary studies.

*Limitations*

The limitation of the presented research is the lack of a representative research samples. The research was conducted on the territory of the Archdiocese of Lublin (one of 15 metropolitan areas in Poland), however the features important for the subject under consideration (the level of religiosity and the level of social involvement) constitute the statistical average for Poland. The collected data constitute sufficient material to outline clear tendencies and characteristics of parishioners as committed members of the parish community. The disadvantage of this research is that it does not take a broader perspective into account—the characteristics of the parish as important factors determining the level of social involvement of parishioners. This is because the analyses presented herein concern only a portion of the research, focusing on indicating those socio-demographic variables that relate to the assessment of parishioners' influence on parish non-religious activities. A better understanding of the potential of the parish as an institution requires further research. The important line of study is undoubtedly the research conducted among priests, which can enhance knowledge about the role of parish capacity as an institutional support for community building.

Another limitation of this study is related to the data collection method. During data collection, results based on participants' own experiences were used, which may cause bias. Participants are often not objective when talking about their own experiences. Respondents may give more socially acceptable answers than truthful ones and may not be able to assess themselves accurately. Nevertheless, when self-report surveys are used correctly, the data can help to obtain a wider range of responses than many other data collection instruments.

**Author Contributions:** Conceptualization, J.P. and M.S.; methodology, J.P. and M.S.; formal analysis, K.J.; investigation, J.P. and M.S.; data curation, J.P., M.S. and K.J.; writing—original draft preparation, J.P., M.S. and K.J.; writing—review and editing J.P., M.S. and K.J.; visualization, M.S. and K.J.;

project administration, J.P. and M.S. All authors have read and agreed to the published version of the manuscript.

**Funding:** This research received no external funding.

**Institutional Review Board Statement:** The study was conducted according to the guidelines of the Declaration of Helsinki and approved by the Ethics Committee of the Institute of Sociological Sciences of the John Paul II Catholic University of Lublin (protocol code: 11/DKE/NS/2021).

**Informed Consent Statement:** Informed consent was obtained from all subjects involved in the study.

**Data Availability Statement:** The data presented in this study are available on request from the corresponding author.

**Conflicts of Interest:** The authors declare no conflict of interest.

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
