# Peer review of "Factors Determining the Involvement in Non-Religious Activities in the Parish: A Cross-Sectional Study of the Catholic Laity"

_religions, doi:10.3390/rel12121097_

Round 1
Reviewer 1 Report
The text is interesting. cognitively valuable, desrving of publication
Author Response
Dear Sir or Madam,
Thank you very much for the positive evaluation of our work.
Reviewer 2 Report
Research on the involvement of the Catholic Laity in the activity of the parish is undoubtedly interesting and fits into the trend of classical research on social engagement (cf. Putnam for example). Experience in North America shows that religious engagement is strongly associated with the civic engagement.
The biggest issue in the current draft is the lack of theoretical positioning of all the empirical information. In other words, the current version is short of demonstrating how this paper is of relevance to readers who are not necessarily familiar with or interested in the polish situation of the Catholic Laity.
The presented article, despite the huge potential and the large number of people surveyed, raises a lot of objections regarding mainly the methodology of the study and the literature used. Generally speaking, the research presented (as indicated at the very end of the text Author(s): line 381-403) has very serious limitations. Alternatively, they could be treated as pilot studies.
Research is neither representative, nor random, nor targeted. Also, the distribution of the sample does not generally correspond to the distribution even in terms of gender in society, despite the focus on socio-demographic variables.
Also, the embedding in the literature indicates numerous shortcomings. It seems biased to rely mainly on data provided by the Catholic Church. It is also incomprehensible to treat the studied area as: "picture" of statustical average "(line 152), to further indicate the differences between other regions in Poland (line 156-167).
It seems that the presented material does not have a sufficient reference to the situation in other countries, which would make it more interesting for readers who are not very familiar with Polish specificity.
In my opinion, the presented article requires numerous corrections before it is allowed to be printed.
First of all, I would like to point out that literature should be presented much more broadly, not necessarily polish. The introduction of more theory and comparison would make it stronger. For example, there is no reference to the theory of social capital, social networks ect.
The Author(s) may also want to improve their articulation of methodology.
The description of the studies should clearly indicate from the outset the shortcomings and limitations in the study.
Ideally, the Author(s) should also explicitly articulate the wide-ranging discussion, not just the local one. In so doing, the revised paper will be able to inspire a broader range of audiences.
The presented material can be treated as a starting point for the further work on the material collected during undoubtedly large and laborious research.
Author Response
Dear Sir or Madam,
Thank you for all comments. We appreciate your suggestions, they allowed us to look at the discussed problems from a wider perspective. In our opinion, they are also valuable in relation to further research that we are conducting. We strongly believe that the suggested corrections and additions have resulted in an improvement of our manuscript. Below we present our responses to the suggestions and recommendations.
Point 1: The biggest issue in the current draft is the lack of theoretical positioning of all the empirical information. In other words, the current version is short of demonstrating how this paper is of relevance to readers who are not necessarily familiar with or interested in the polish situation of the Catholic Laity.
It seems that the presented material does not have a sufficient reference to the situation in other countries, which would make it more interesting for readers who are not very familiar with Polish specificity.
Response: We have made appropriate revisions to the manuscript. We have significantly supplemented the text based on the literature on parish involvement. We discussed themes that are universal, independent of strictly national conditions. For example, the literature used in the article referring to the situation in the USA or Australia clearly focuses the researchers' attention on activities for strengthening the sphere of religious life and building religious social capital in parishes. It should be emphasized that this is a frequently explored area, and thus quite well recognized by sociologists of religion. At the same time, however, we wanted to draw attention to the under-recognized mechanism of parishioners' activity directed at non-religious needs, which some members of parish communities are unable, for various reasons, to satisfy independently - outside the parish community (lines 347-351; 388-437).
Point 2: The presented article, despite the huge potential and the large number of people surveyed, raises a lot of objections regarding mainly the methodology of the study and the literature used. Generally speaking, the research presented (as indicated at the very end of the text Author(s): line 381-403) has very serious limitations. Alternatively, they could be treated as pilot studies.
Resposne: We agree with the Reviewer. We are aware of the limitations of our research. As suggested by the Reviewer, information about the pilot study was added in the introduction of the text (line 166-168; lines 231-236).
Point 3: Research is neither representative, nor random, nor targeted. Also, the distribution of the sample does not generally correspond to the distribution even in terms of gender in society, despite the focus on socio-demographic variables. The description of the studies should clearly indicate from the outset the shortcomings and limitations in the study. The Author(s) may also want to improve their articulation of methodology.
Response: The analyses in this article are part of a larger study on the perception of parishes as a source of social support. Our intention was to examine certain categories of people who may need this type of support. Therefore, the research sample was specific. The criteria for selecting respondents included the following categories of people or groups: families with children, single parents, seniors (60+), singles, dependent people (including those with disabilities). However, in this article we focus on only one aspect - the involvement of parishioners in non-religious activities in the parish. We agree that the survey is neither representative, nor random, nor targeted, as we have emphasized in the manuscript. At the same time, we believe that the results can be used to outline trends and characteristics of parishioners as engaged members of the parish community in non-religious activities. The description of the survey is supplemented by a clear indication of the lack of representativeness of the research sample (line 231-236).
Point 4: I would like to point out that literature should be presented much more broadly, not necessarily polish. The introduction of more theory and comparison would make it stronger. For example, there is no reference to the theory of social capital, social networks ect.
Response: Thank you for your valuable comment. We have made an addition to the text. Indeed, it should be acknowledged that the supplementation of the theoretical basis of the text with issues related to social capital, especially religious social capital, has significantly broadened the analysis of the research problem. This involved the inclusion of new bibliographic items presenting research on various elements of social capital of religious structures in different cultural contexts of several countries (parishes in the United States, Australia, or Great Britain). This underlined the specificity of the presented research, which emphasizes especially the non-religious dimension of parish activity and the activity of parishioners. A topic that is not very widely studied empirically in sociological studies on the transformation of religiosity in contemporary religious communities (lines 133-168).
Point 5: Ideally, the Author(s) should also explicitly articulate the wide-ranging discussion, not just the local one. In so doing, the revised paper will be able to inspire a broader range of audiences.
Response: Thank you for your valuable comment. We have discussed the non-religious involvement of parishioners without particularly emphasizing localism. It is known that in many countries (e.g. USA and Canada) there is a market for faith-based social services, i.e. entities associated with faith communities. These entities operate commercially, in a formalized manner, in the social sphere for the benefit of society. Against this background, therefore, the spontaneous aid activity of parishioners, which we studied, shows the real effects of creating and implementing social capital in communities. This may be a particularly interesting phenomenon for the reader from different cultural areas (lines 388-437).
Reviewer 3 Report
This is a well written article, depicting what appears to be solid research. I found what is here very interesting. I would have liked to see more analysis of the findings. I understand that sociologists prefer to present findings and leave the analysis to others but a bit more explanation of the implications of the findings would have been welcome.
You also mention the four dimensions of social involvement, however, I don't see where these were tested or discussed, per se. That would have been most interesting. Perhaps in another article??
Author Response
Dear Sir or Madam,
We would like to thank you for taking the time and effort necessary to review the manuscript. They allowed us to look at the discussed problems from a wider perspective and are also valuable in relation to further research that we are conducting. The constructive comments and suggestions helped us to substantially improve our manuscript. Below we present our responses to the suggestions and recommendations.
Point 1: I would have liked to see more analysis of the findings.
Response: Thank you for your comment. We have broadened the conclusions, as suggested by the Reviewer (lines lines 347-351; 388-437).
Point 2: You also mention the four dimensions of social involvement, however, I don't see where these were tested or discussed, per se.
Response: Out of the four dimensions of the model of social participation/social involvement, the text of the article directly relates to two of them: a) the second dimension concerning the role of axiological content (especially religious, moral, social values ​​and focus on helping other people) constituting the basis of non-religious activity of the respondents. Some convergent single indicators defining the social and moral attitudes of the parishioners we surveyed can be mentioned, such as, for example, voluntary commitment, orientation towards the common good of the community or experiencing the values ​​of the community b) the third dimension concerning the sense of influence on the reality and effectiveness in the actions of individuals. The empirical data of own research was discussed in the part relating to social characteristics such as: a stronger sense of influence on the non-religious life of the parish, a higher sense of agency and participation in decision-making of parishioners who express high interest in parish affairs and who hold positions in parish organizations. The first dimension of the model (subjective justification of participation) and the fourth (experience of participationwere only mentioned and not articulated in the content of the article, because both topics - in our opinion - require the use of qualitative rather than quantitative research methods for proper knowledge; we plan to include these issues in a future study.
Round 2
Reviewer 2 Report
Thank you Author(s) for the corrections made - I am satisfied with the supplement on social capital.
Nevertheless, a few issues could still be added. There is still no attempt to indicate what the academic reader from a country other than Poland can get out of reading. It is difficult to say, for example, whether belonging to a parish is a factor causing involvement in social activities, or whether they are people with such predispositions. But still, the biggest issue in the current draft is the lack of theoretical positioning of all the empirical information.
Point 1// Response
“At the same time, however, we wanted to draw attention to the under-recognized mechanism of parishioners' activity directed at non-religious needs, which some members of parish communities are unable, for various reasons, to satisfy independently - outside the parish community (lines 347-351; 388-437).”
This mechanism is the need for affiliation and the creation of communities based on other ties than primordial.
Point 3//Response
In my opinion, the scientific article cannot be based on the belief that they can show a trend. This is still a pilot study and should be treated as such.
It would be good if the author tried to try to show what the author brings new and interesting, more to indicate, to emphasize the inspiration for subsequent research.
Author Response
Dear Sir or Madam,
Thank you for all comments. We appreciate your suggestions, they allowed us to look at the discussed problems from a wider perspective. In our opinion, they are also valuable in relation to further research that we are conducting. We strongly believe that the suggested corrections and additions have resulted in an improvement of our manuscript. Below we present our responses to the suggestions and recommendations.
Point 1
“At the same time, however, we wanted to draw attention to the under-recognized mechanism of parishioners' activity directed at non-religious needs, which some members of parish communities are unable, for various reasons, to satisfy independently - outside the parish community (lines 347-351; 388-437).”
This mechanism is the need for affiliation and the creation of communities based on other ties than primordial.
Response: We appreciate your opinion. The manuscript adds (lines no. 475-490) the explanation of the basis of the action mechanism for addressing the non-religious needs of part of the parish community. Engagement of this kind stems first of all from the essence of the parish as a religious-social structure and from the nature of the social bonds within it, especially from the differentiation of objective and subjective bonds. The objective bond defines the belonging of the faithful to the parish as an institution due to their place of residence, while the subjective one is based on the community of values, norms, symbols, identification with the parish community, which - in the sense of a moral bond based on trust and solidarity - may generate the need for causal action and influence on the environment through charitable activities or other forms of social action. This mechanism requires further, deeper insight in sociological and psychological research.
Point 2
In my opinion, the scientific article cannot be based on the belief that they can show a trend. This is still a pilot study and should be treated as such.
It would be good if the author tried to try to show what the author brings new and interesting, more to indicate, to emphasize the inspiration for subsequent research.
Response: We have expanded the discussion (lines no. 491-505). Additionaly, we have identified areas for further research on non-religious involvement (lines 516-528).
We have added new references.
